# Evaluation of SSTR2 Expression in SI-NETs and Relation to Overall Survival after PRRT

**DOI:** 10.3390/cancers13092035

**Published:** 2021-04-23

**Authors:** Anna-Karin Elf, Viktor Johanson, Ida Marin, Anders Bergström, Ola Nilsson, Johanna Svensson, Bo Wängberg, Peter Bernhardt, Erik Elias

**Affiliations:** 1Department of Surgery, Institute of Clinical Sciences, Sahlgrenska Academy, University of Gothenburg, 405 30 Gothenburg, Sweden; viktor.johanson@medfak.gu.se (V.J.); bo.wangberg@surgery.gu.se (B.W.); erik.elias@vgregion.se (E.E.); 2Department of Endocrine Surgery, Sahlgrenska University Hospital, 405 30 Gothenburg, Sweden; 3Department of Radiation Physics, Institute of Clinical Sciences, Sahlgrenska Academy, University of Gothenburg, 405 30 Gothenburg, Sweden; ida.marin@vgregion.se (I.M.); peter.bernhardt@radfys.gu.se (P.B.); 4Department of Pathology, Sahlgrenska University Hospital, 405 30 Gothenburg, Sweden; anders.c.bergstrom@vgregion.se; 5Department of Pathology and Genetics, Institute of Biomedicine, Sahlgrenska Academy, University of Gothenburg, 405 30 Gothenburg, Sweden; ola.nilsson@llcr.med.gu.se; 6Department of Oncology, Sahlgrenska University Hospital, 405 30 Gothenburg, Sweden; johanna.b.svensson@vgregion.se; 7Department of Oncology, Institute of Clinical Sciences, Sahlgrenska Academy, University of Gothenburg, 405 30 Gothenburg, Sweden; 8Department of Medical Physics and Biomedical Engineering, Sahlgrenska University Hospital, 405 30 Gothenburg, Sweden

**Keywords:** small intestinal neuroendocrine tumor (SI-NET), peptide receptor radionuclide therapy (PRRT), somatostatin receptor expression, overall survival

## Abstract

**Simple Summary:**

Small intestinal neuroendocrine tumors (SI-NETs) are slow growing tumors expressing somatostatin receptors (SSTR), which are targeted in diagnostic and therapeutic methods. A fairly new treatment that targets SSTR2 is peptide receptor radionuclide therapy (PRRT), which prolongs survival for patients with metastasized NETs. However, the treatment is costly, and the effect is variable. Therefore, finding predictors for treatment response is warranted. The aim of this retrospective study was to immunohistochemically analyze the SSTR2 expression in SI-NETs, using a previously constructed tissue microarray, and to investigate if a high SSTR2 expression was correlated to overall survival (OS). Among 42 patients that had received PRRT, 10 had at least one tumor with low SSTR2 expression. The patients were grouped according to the SSTR2 expression (“High SSTR2” and “Low SSTR2”) in previously resected tumors. In contrast to the hypothesis of the study, patients with low SSTR2 expression had significantly longer OS after PRRT, compared with patients with high SSTR2 expression. Hence, the study suggests that low SSTR2 expression in resected tumors should not exclude SI-NET patients from receiving PRRT.

**Abstract:**

(1) Purpose: Small intestinal neuroendocrine tumors (SI-NETs) often present with distant metastases at diagnosis. Peptide receptor radionuclide therapy (PRRT) with radiolabeled somatostatin analogues is a systemic treatment that increases overall survival (OS) in SI-NET patients with stage IV disease. However, the treatment response after PRRT, which targets somatostatin receptor 2 (SSTR2), is variable and predictive factors have not been established. This exploratory study aims to evaluate if SSTR2 expression in SI-NETs could be used to predict OS after PRRT treatment. (2) Methods: Using a previously constructed Tissue Micro Array (TMA) we identified tissue samples from 42 patients that had received PRRT treatment during 2006–2017 at Sahlgrenska University hospital. Immunohistochemical expression of SSTR2, Ki-67 and neuroendocrine markers synaptophysin and Chromogranin A (CgA) were assessed. A retrospective estimation of ^177^Lu-DOTATATE uptake in 33 patients was performed. Data regarding OS and non-surgical treatment after PRRT were collected. Another subgroup of 34 patients with paired samples from 3 tumor sites (primary tumor, lymph node and liver metastases) was identified in the TMA. The SSTR2 expression was assessed in corresponding tissue samples (*n* = 102). (3) Results: The patients were grouped into Low SSTR2 or High SSTR2 groups based upon on levels of SSTR2 expression. There was no significant difference in ^177^Lu-DOTATATE uptake between the groups. The patients in the Low SSTR2 group had significantly longer OS after PRRT than the patients in the High SSTR2 group (*p* = 0.049). PRRT treated patients with low SSTR2 expression received less additional treatment compared with patients with high SSTR2 expression. SSTR2 expression did not vary between tumor sites but correlated within patients. (4) Conclusion: The results from the present study suggest that retrospective evaluation of SSTR2 expression in resected tumors cannot be used to predict OS after PRRT.

## 1. Introduction

Neuroendocrine tumors (NETs), originating from enterochromaffin cells in small intestinal mucosa (SI-NETs), are the most common small intestinal neoplasms with a reported incidence of 1–5/100,000 [1,2,3].

Patients with SI-NETs are often diagnosed with synchronous regional or distant metastases (WHO stage III-IV) [4]. The only potentially curative treatment is radical surgical resection. Although a large proportion of SI-NET patients with stage III-IV disease cannot receive curative treatment, they often undergo surgery of the primary tumor and mesenterial lymph nodes to avoid or resolve a bowel obstruction. Patients with disseminated disease still have a long expected overall survival (OS). The 5 year-OS for patients with stage III disease is more than 90%, and for stage IV disease it is 60–80%, which possibly reflects the generally low proliferative rate in SI-NETs [1,5,6]. As a consequence, a large number of SI-NET patients will over time receive several treatments beyond surgical resection.

SI-NETs generally express the somatostatin receptor subtype 2 (SSTR2) [7]. Treatment with somatostatin analogues (SSA) reduces hormone secretion and has an antitumoral effect [8,9]. SSTR2 expression is also used clinically for molecular imaging, by scintigraphy (^111^In-octreotide) or the more recently developed SSTR PET/CT (^68^Ga-DOTATOC/TATE-PET). The SSTR2 expression can further be exploited therapeutically by targeting NETs with radiolabeled SSA (^177^Lu-DOTATATE or ^177^Lu-DOTATOC), i.e., peptide receptor mediated radionuclide therapy (PRRT) [10].

PRRT in combination with SSA has recently been evaluated in a randomized clinical trial, in which PRRT and SSA increased time to progression and OS compared to SSA alone [11]. However, even though patients are selected for PRRT by evaluation of SSA uptake with SSTR imaging, the objective response of PRRT is still highly variable [12] and, especially for SI-NETs, difficult to assess due to the slow tumor progression rate [13]. Predictors of long-term outcome after PRRT are lacking. Radiological treatment response after PRRT has been proposed as a prognostic marker for prolonged survival in a cohort of diverse NETs; however, if there is a similar correlation for SI-NETs as a group, it still remains unclear [14]. There are promising results from studies on new blood biomarkers as predictors for RECIST response, but it has not yet been clarified if these could serve as predictive tools for long-term results after PRRT [15].

SSTR2 expression can be assessed with immunohistochemistry (IHC) in tumor samples. The method is semi-quantitative and can only be used to quantify relative expression among samples. In a clinical setting, all SI-NET metastases are not biopsied prior to PRRT and therefore it is not possible to quantify the SSTR2 expression in all PRRT treated lesions. It has not previously been determined if SSTR2 protein expression in lesions treated with PRRT influences long-term outcome.

This pilot study was designed to explore the overall hypothesis that SSTR2 expression in resected tissue could be used to predict OS after subsequent PRRT. This hypothesis is based on the following assumptions: (i) SSTR2 expression in SI-NET correlates among a patient’s lesions, i.e., the presence of one tumor with a low SSTR2 expression indicates that other lesions also could have a low SSTR2 expression. (ii) Low SSTR2 expression in resected tissue could affect subsequent SSA uptake and thereby (iii) affect efficacy of later PRRT treatment and influence OS.

In order to test these assumptions, we identified a cohort of PRRT treated patients with samples present on a tissue micro array (TMA) which enables a relative quantification of SSTR2 expression among this patient cohort.

## 2. Materials and Methods

### 2.1. Tumour Tissue Samples

A TMA block was assembled as previously described [16]. Briefly, all patients who underwent surgery for SI-NET at Sahlgrenska University Hospital from 1986 to 2013 were included in a TMA. Formalin-fixed and paraffin-embedded tumor tissue from this cohort was retrieved from the Department of Clinical Pathology and Genetics, Sahlgrenska University Hospital, Gothenburg, Sweden. The diagnosis was confirmed by reviewing hematoxylin and eosin-stained sections and IHC stainings. Sufficient tumor material for construction of the tissue microarray was available from 412 patients. A total of 8 recipient blocks were constructed, derived from 846 tumors. The quality of the constructed TMA was evaluated on hematoxylin and eosin-stained sections.

### 2.2. Immunohistochemistry and Scoring

Sections from the TMA blocks were placed on coated glass slides and were subjected to antigen retrieval using EnVision FLEX Target Retrieval Solution (high pH) in a Dako PT-Link. IHC staining was performed in a Dako Autostainer Link using EnVision FLEX according to the manufacturer’s instructions (DakoCytomation, Glostrup Denmark).

The following primary antibodies were used: anti-SSTR2a (clone UMB1; cat no. 134,152 Abcam, Cambridge, UK), anti-chromogranin A (MAB319; Chemicon/Merck, Massachusetts, USA), anti-synaptophysin (SY38, M0776; Dako, Glostrup Denmark) and anti-Ki67 (clone MIB-1; cat no M7240; Dako, Glostrup Denmark); positive and negative controls were included in each run. The fraction of Ki67 positive cells was assessed by manually counting 500–2000 tumor cells per sample in full section slides corresponding to the TMA core biopsy [17].

Stained TMA slides were scanned using Leica SCN 4000 at × 40 magnification. The scoring system was based on the immunoreactive scoring (IRS) method, as previously described by Specht et al. [18]. In short, a score for staining intensity between 0–3 was determined using all 846 tumors. A scoring (1–4) of positively stained cells was also performed according to the following: 1 ≤ 10% positive cells, 2 = 10–50% positive cells, 3 = 51–80% positive cells and 4 ≥ 80% positive cells. These two scores were multiplied for a combined score of 0–12, which was then divided into separate groups (score 0–1 = group 0, score 2–3 = group 1, score 4–8 = group 2 and score 9–12 = group 3).

When we applied this method in our samples, we found a consistently homogenous expression pattern with over 80% stained cells in all our samples with the exception of 4 negative samples (i.e., score 0). Therefore, staining intensity was the primary determinant for our final score (0–3). It should be noted that the observed homogenous staining intensity is a possible effect of using a TMA that consists of small samples from a larger paraffin embedded sample (Appendix A). Thus, intra-tumor heterogeneity regarding SSTR2 expression could not be assessed. Regarding subcellular staining pattern, there was a strong membranous staining pattern and a slightly weaker cytoplasmatic staining. In general, the intensity of the membranous staining and the cytoplasmatic staining within a sample appeared to be strongly correlated. Consequently, samples with the most intense membranous staining also had the most intense cytoplasmatic staining and as a result, a higher score (2 or 3) in overall intensity. The SSTR2 expression in the entire TMA was initially scored by a board-certified pathologist (O.N.) Samples corresponding to patients included in the present study were reassessed by two blinded observers (E.E. and A.-K.E.). Two cases differed in the SSTR2 score between the 3 observers and for these cases the score that 2 out of the 3 observers agreed upon was chosen. Synaptophysin and Chromogranin A (CgA) were scored by a single observer (E.E.).

### 2.3. Patients and Clinical Characteristics

Among the specimens on the TMA block we identified samples from 44 patients treated with PRRT during the years 2006–2017 at Sahlgrenska University Hospital. Flowchart, indications and exclusion criteria for PRRT are presented in Figure 1.

Clinical characteristics of patients are presented in Table 1. For comparisons related to SSTR2 expression, the patients were grouped according to SSTR2 score in previously resected tumor tissue. We argued that the presence of at least one tumor lesion with low SSTR2 expression implied a risk of having more lesions with low SSTR2 expression. Therefore, patients with at least one (1) tumor sample with scores 0 or 1 were assigned to the “Low SSTR2” group and the remaining patients (all samples scored 2–3) were assigned to the “High SSTR2” group. This stratification resulted in 10 patients in the Low SSTR2 group and 32 patients in the High SSTR2 group. Clinical data regarding OS and other treatments were obtained. Two PRRT treated patients died before they could complete the intended PRRT treatment and were therefore excluded from the survival analysis. When Ki-67 was determined for a patient represented by more than one sample, the highest Ki-67 was used.

### 2.4. Activity Concentration in Tumors

In PRRT treated patients, an estimation of the uptake of radionuclide (^177^Lu) was done by measuring the activity concentration in the tumors (Figure 2). In SPECT/CT images acquired 24 h after the first PRRT treatment, reconstructions were done with the recently developed Monte Carlo based ordered subset expectation maximization algorithm SARec [19]. Tumors were identified by visual inspection and the three tumors containing the highest maximum voxel values in each patient were chosen for assessment. Activity concentration calculation was done by dividing the maximum voxel value with SPECT sensitivity and mass of the tissue represented by the voxel.

### 2.5. Statistical Analysis

For all statistical analysis of data generated from IHC scoring, non-parametric tests were used. For comparisons between 2 groups the Mann–Whitney U test was used. For comparisons between 3 or more groups the Kruskal–Wallis test was used. Spearman rank correlation was performed for correlation analyses. For survival curve comparisons the Mantel–Cox log-rank test was used. A level of significance was set to *p* < 0.05 in all tests. Statistical analyses and graphic design were performed in Prism 9 for MacOS (GraphPad software).

## 3. Results

### 3.1. SSTR2 Scoring and Distribution of SSTR2 Expression Among Samples

Representative images of SSTR2 expression score and distribution of SSTR2 score among lesions are presented in Figure 3. In 95 samples from the 44 PRRT treated patients, 16 samples (16.8%) had no or low STTR2 expression (score 0 or 1). The majority of samples (*n* = 79) had medium or high SSTR2 expression (score 2 or 3). To assess the stability of antigen preservation over time we performed a correlation analysis between time of surgery and SSTR2 score for all samples on the TMA (Appendix A). There was no significant correlation, which argued against age of samples as a factor determining SSTR2 score.

In order to determine SSTR2 expression in tumors in different sites in a patient we identified 34 patients who had samples from both a primary tumor, lymph node metastasis and liver metastasis on the TMA. Samples from these patients were only used to compare SSTR2 expression levels and were not correlated with clinical data. The SSTR2 expression levels did not vary significantly between primary, lymph node or liver metastases (average score primary tumor 2.18 vs. average score lymph node metastases 2.03 vs. average score liver metastases 2.24, *p* = 0.52) (Appendix A).

We then aimed to determine if the SSTR2 score correlated among lesions within a single patient. Three groups of primary tumors (score 1–3, no primary tumor had score 0) were established and SSTR2 expression in matched metastases was assessed (Figure 4). The SSTR2 score correlated in all tumor samples within a patient, when sorted according to SSTR2 score in the primary tumor. Thus, the SSTR2 expression in tumor samples did not significantly change in metastases.

### 3.2. SSTR2 Expression Does Not Correlate with Synaptophysin or CgA Expression

We also assessed if SSTR2 expression was associated with the IHC expression of established SI-NET markers synaptophysin and CgA. A subset of samples (*n* = 49) from PRRT treated patients were divided into groups based on SSTR2 expression, and synaptophysin and CgA expression for each sample was assessed. IHC expression of synaptophysin and CgA was consistent among samples, regardless of SSTR2 expression (Figure 5).

### 3.3. SSTR2 Expression and Proliferation Rate

To determine the association between SSTR2 and proliferation rate, Ki-67 was assessed in 66 samples present on the TMA. These samples from PPRT treated patients (Lower box in Figure 1) were then sorted according to SSTR2 expression. The samples included tissue from a primary tumor, lymph node or liver metastases and the patients were represented by 1–3 samples. All samples with high Ki-67 also had a high SSTR2 expression (score 2–3); however, there was no statistically significant difference between the groups (*p* = 0.25) (Figure 6a). The samples were then sorted according to the patient groups “High SSTR2” (*n* = 32) and “Low SSTR2” (*n* = 10). For patients represented by more than one sample, the highest Ki-67 was used. In the Low SSTR2 group, all patients but one had WHO grade 1 tumors (Ki-67 < 3%). In the High SSTR2 group more than one third of the patients (*n* = 11) had WHO grade 2 tumors (Ki-67 3–20%). However, there was no statistically significant difference between the groups (*p* = 0.10) (Figure 6b).

### 3.4. SSTR2 Expression and Activity Concentration

In PRRT treated patients, an estimation of the uptake of radionuclide was done by measuring the activity concentration in 3 tumors per patient (in two patients only 2 tumors were measured). Measurements were possible in 97 tumors from 33 patients. Results showed a large variability between treated patients; however, there was a consistency among the tumors within patients. The patients were grouped according to SSTR2 expression in the previously resected tumors from the TMA. This resulted in 79 tumors from 27 patients with high and 18 tumors from 6 patients with low SSTR2 expressing tumors, respectively (Figure 7). There was no significant difference in activity concentration between the groups (*p* = 0.99).

### 3.5. SSTR2 Expression and Long-Term Outcome

After the first PRRT, all 42 patients were followed for 69 months (mean; range 5–161 months). At the last follow-up, 15 patients were still alive (7 were in the Low SSTR2 group). The mean follow-up time in the High and the Low SSTR2 groups was 57 and 81 months after PRRT, respectively. Kaplan–Meier curves depicting OS based on SSTR2 expression in these two patient cohorts showed a statistically significant difference between the groups (*p* = 0.049) (Figure 8).

### 3.6. SSTR2 Expression and Treatment Patterns

In accordance with clinical practice, all patients received SSA in the form of short-acting octreotide 100 µg × 4 peri-operatively. At initiation of PRRT treatment, patients had progressive and/or symptomatic stage IV disease, hence at this timepoint all patients except 5 had SSA treatment. However, we did not have records on how long they had received SSA prior to PRRT. Most patients received long-acting formulas administered once every 28 days: Lanreotide 60–120 mg (*n* = 21 patients) or Octreotide LAR 20–30 mg (*n* = 8 patients). Eight patients used short-acting Octreotide 100–200 µg 1–3 times daily.

After PRRT treatment, patients continued SSA, and were offered additional treatments in case of tumor progression. Eight of the 32 patients (25%) with high SSTR2 expression received additional treatment after initial PRRT. None of the patients (0/10) with low SSTR2 expression received additional treatment during follow-up (Table 2).

## 4. Discussion

This pilot study was designed to test the following assumptions: (i) SSTR2 expression in SI-NETs correlates among a patient’s lesions, i.e., the presence of one tumor with a low SSTR2 expression indicates that the patient could have other lesions with a low SSTR2 expression. (ii) A low SSTR2 expression could affect SSA uptake and thereby (iii) affect efficacy of PRRT treatment and influence OS.

In the present study, we found varying SSTR2 expression in tumor lesions from SI-NET patients, which is concordant with other studies [20]. To address the assumption that SSTR2 correlates among a patient’s lesions, we identified paired samples from primary tumor, lymph node and liver metastases from 34 patients represented on the TMA. The SSTR2 expression did not differ based on tumor location. Instead, when studying the intra-patient variability, we found that lesions at different locations within a patient had significantly similar SSTR2 expression. Although there was some variance regarding the correlation, our interpretation is that IHC assessment of SSTR2 expression in resected tumors could be representative for the remaining metastases, later targeted with PRRT therapy.

There are some methodological considerations to address. Several different methods of scoring for evaluating SSTR2 immunohistochemically have been proposed. Körner et al. compared SSTR2 expression, using the then newly developed UMB-1 antibody, with SSTR autoradiography and found that IHC staining of >10% of tumor cells corresponded to SSTR levels high enough for clinical applications [21]. The use of a scoring system based on both staining intensity and percentage of stained cells has been advocated by some authors [22,23], while others have emphasized the importance of the subcellular localization [24]. Staining patterns can be influenced by a specific IHC methodology, representing a challenge when comparing results from different studies. We adapted the IRS score to quantify SSTR2 expression in our samples. In general, our samples had a homogenous staining pattern and all cells within a section exhibited the same staining pattern. We, therefore, chose to score the samples solely based upon staining intensity. It can be noted that although we experienced a general concordance between overall staining intensity and membranous staining, we did not specifically assess only membranous staining and it is possible that this could have influenced the results. A faint SSTR2 staining could be caused by suboptimal tissue fixation with impaired antigen preservation. As the expression of synaptophysin and CgA was not associated with varying SSTR2 expression, we concluded that the variation in SSTR2 expression was not caused by impaired antigen preservation. Other important methodological considerations include the use of a TMA, which represents a much smaller tissue sample than in commonly used full slides (Appendix A). Consequently, heterogeneity in the expression of SSTR2 in individual samples could be missed.

To address the assumption that a low SSTR2 expression could affect SSA uptake and thereby affect the efficacy of PRRT and influence OS we identified 42 patients that had received a complete PRRT treatment cycle during 2006–2017 at Sahlgrenska University Hospital. These 42 patients had at least one sample present on the TMA allowing for IHC analyses. Ten of forty-two patients had at least one tissue sample with a low SSTR2 expression, and these patients were grouped into a “Low SSTR2” expression group while all remaining patients were assigned to the “High SSTR2” group. The SSA uptake, measured by activity concentration after PRRT, was similar in all of a patients’ lesions and there was no difference in activity concentration between the groups. We chose to assess activity uptake in tumors by SPECT/CT in relation to the first PRRT treatment, as this investigation enables direct evaluation and should correspond well with pre-treatment SSTR imaging [25]. Our data do not show that a low SSTR2 expression predicts a lesser activity uptake, which is concordant with some studies [23] but in contrast to others, where a correlation was found between SSTR2 expression and activity uptake on SSTR imaging [20,26,27]. However, the previous studies include NETs from various origins and both low- and high-grade tumors, which might explain this discrepancy. The results indicate that the level of SSTR2 expression in surgically resected tumors cannot be used to predict radionuclide uptake after PRRT.

The present study showed a significantly prolonged survival among patients with low SSTR2 expressing tumors. This is in contrast to other studies, where high SSTR2 expression was associated with prolonged survival [23,28,29]. These contradictory findings could be due to patient selection bias or different types of NETs. Compared with the studies above, the present study included only a minority of tumor samples with low SSTR2 expression, possibly introducing a risk of skewed results. However, the reports above include heterogenous patient cohorts and inform insufficiently about the treatments given, making it unclear whether the results show SSTR2 as a prognostic marker for a more indolent disease course or a predictive marker for the (undefined) therapies given. A known independent predictor for OS is the proliferative marker Ki-67 [30,31]. In the present study, Ki-67 was generally low in the tumor samples, which is expected in a SI-NET population, and there was no difference between Ki-67 in the high and low SSTR2 groups. Therefore, we concluded that the survival differences seen between the groups could not be explained by Ki-67. It should be noted that there are other SSTR expressed in SI-NET that could potentially influence PRRT outcome. In an article by Qian et al. [23], the expression of SSTR subtypes was generally correlated among each other but only SSTR2 expression was predictive of clinical outcome, which influenced our decision to only assess SSTR2 in the present study.

The overall hypothesis of this study was that SSTR2 expression in resected tissue could be used to predict OS after subsequent PRRT. The rationale behind the hypothesis was to identify predictive factors for PRRT that could be easily implemented in a clinical setting. However, the study has several limitations. The limited number of patients increases the risk for type 2 errors. Furthermore, the tumor samples used were obtained some time before PRRT, which is an important factor to consider when interpreting the results. The median time between sample collection (i.e., surgery) and initial treatment was 60 months (Appendix A). During this time SSTR2 expression could have changed in the remaining lesions. Therefore, the present results cannot determine if current SSTR2 expression affects activity uptake or efficacy of PRRT treatment; however, they suggest that retrospective evaluation of SSTR2 should not influence the decision to offer a patient PRRT.

A strength of the study is the long follow-up, which given the natural course of SI-NETs, is required for evaluating long-term outcome such as survival. Another strength is the homogenous patient cohort, containing only SI-NETs grade 1 and 2, with a similar treatment protocol and prognosis. In the literature, reports evaluating PRRT often include cohorts with tumors of diverse origins and grades, making it more difficult to extrapolate these results [20,26,27]. Furthermore, the use of a TMA facilitates IHC staining under identical conditions for all samples, enabling relative quantification of protein expression in the tumor samples [32].

## 5. Conclusions

In order to optimize individualized systemic therapy for SI-NETs there is a need for predictive markers for PRRT. Altogether, the results from the present study suggest that retrospective evaluation of SSTR2 expression cannot be used to predict response to PRRT.

## Figures and Tables

**Figure 1 cancers-13-02035-f001:**
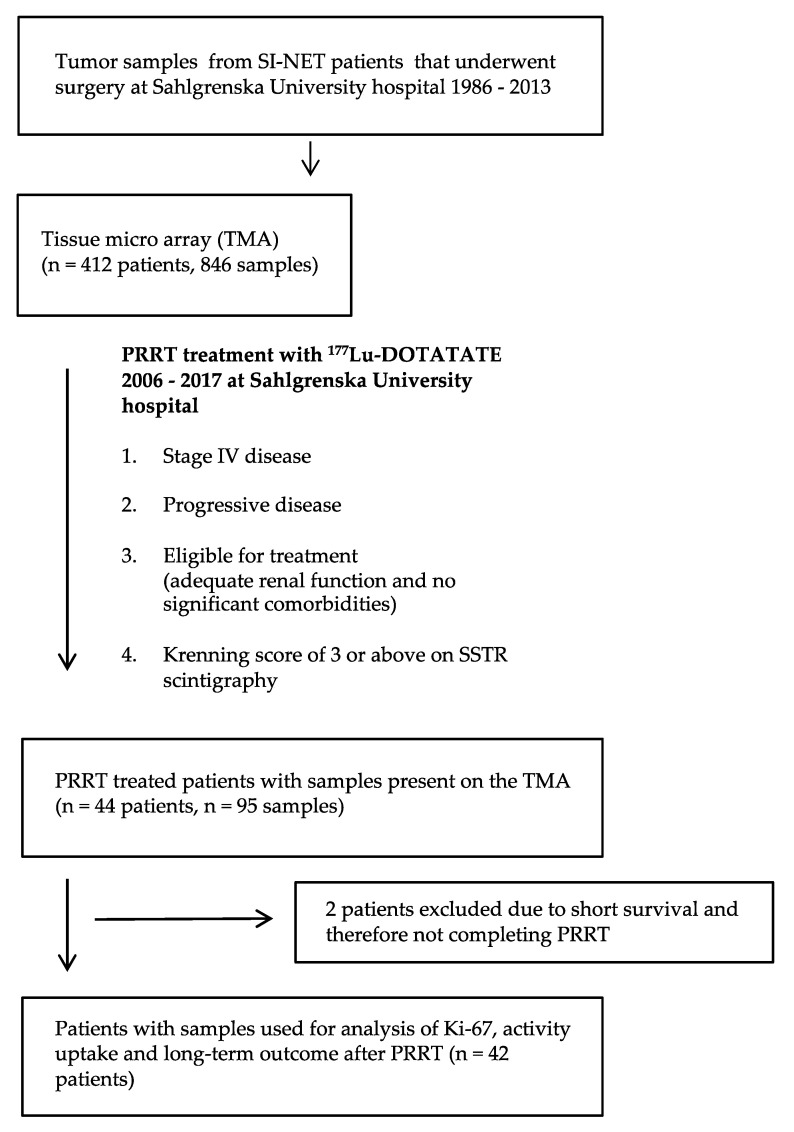
Flowchart describing identification of patient cohort and corresponding analysis.

**Figure 2 cancers-13-02035-f002:**
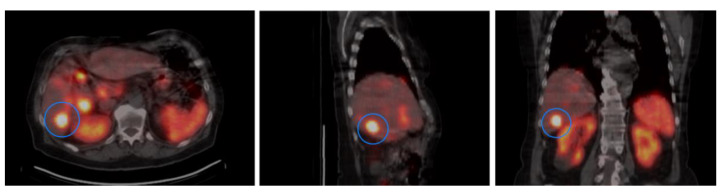
The activity concentration was calculated by using data from SPECT/CT imaging 24 h after the first PRRT treatment. Figure shows representative imaging in trans-axial, sagittal and coronal planes. The tumor lesion with the highest uptake is indicated.

**Figure 3 cancers-13-02035-f003:**
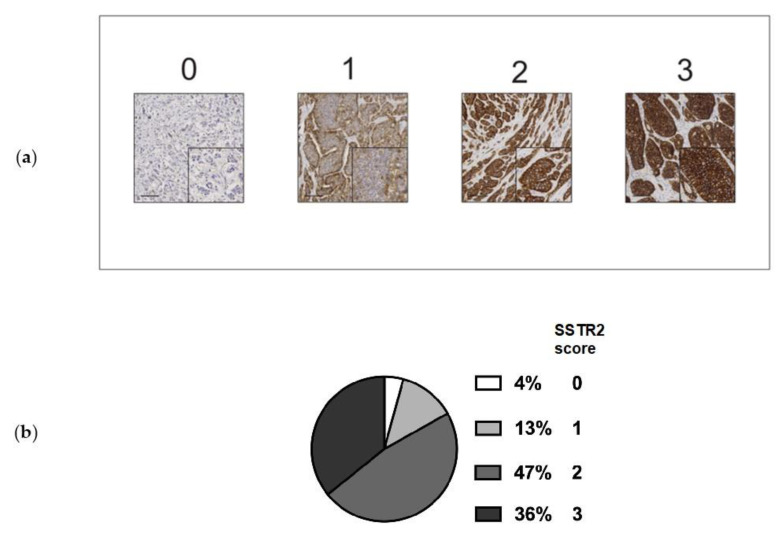
(**a**) Representative images of SSTR2 scoring 0–3. Size bar = 100 µm. Insert shows magnification. (**b**) Distribution of SSTR2 score in tumor samples (*n* = 95) from PRRT treated patients.

**Figure 4 cancers-13-02035-f004:**
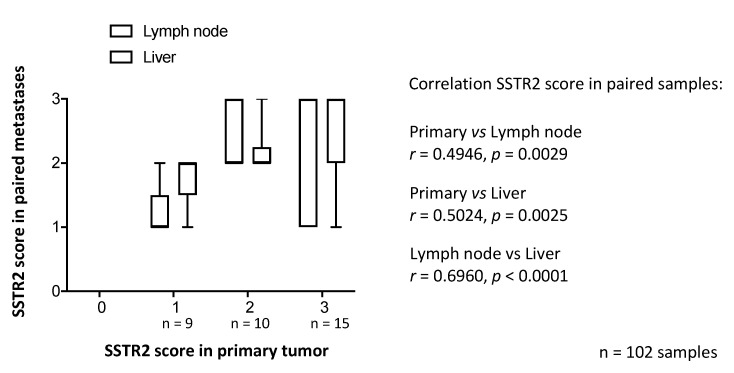
SSTR2 score in primary tumor compared with corresponding metastases. Samples from both primary tumor, lymph node and liver metastases from 34 patients were analyzed (3 samples each from 34 patients, in total 102 samples). When samples from metastases were sorted according to the SSTR2 score in the corresponding primary tumor, a consistency in SSTR2 score between primary tumors and metastases was observed. The figure illustrates a “box and whiskers” plot: boxes show 25th to 75th percentile and whiskers indicate range. Spearman rank correlation.

**Figure 5 cancers-13-02035-f005:**
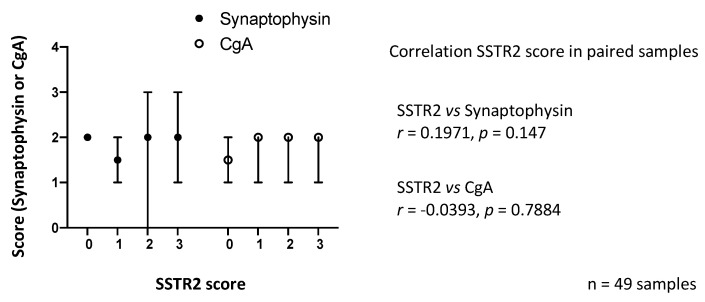
Synaptophysin (syn) and Chromogranin A (CgA) staining intensity compared with SSTR2 expression in a subset of samples (*n* = 49). Staining intensity was consistent among samples regardless of SSTR2 expression level. Dots and bars indicate medians and ranges, respectively. Spearman rank correlation.

**Figure 6 cancers-13-02035-f006:**
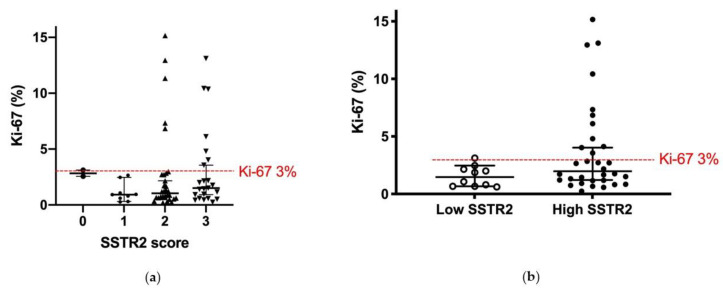
Ki-67 and SSTR2 expression of a subgroup of samples (*n* = 66) in the TMA. Ki-67 was assessed in full section slides corresponding to the TMA core biopsy. (**a**) Most samples (*n* = 53) had Ki-67 < 3% (WHO grade 1). All but one sample (SSTR2 score 0) with grade 2 tumors (Ki-67 3–20%) had consistently high SSTR2 expression (score 2–3). There was no significant difference between the groups (*p* = 0.25; Kruskal–Wallis test); (**b**) When sorting the samples according to patient groups High SSTR2 (*n* = 32) and Low SSTR2 (*n* = 10), almost all patients with grade 2 tumors also had high SSTR2 expressing tumors. For patients with more than one sample, the highest Ki-67 was used. There was not a significant difference between the groups (*p* = 0.10; Mann–Whitney U test). Bars show median and 95% CI. Red line shows 3% threshold between grade 1 and grade 2.

**Figure 7 cancers-13-02035-f007:**
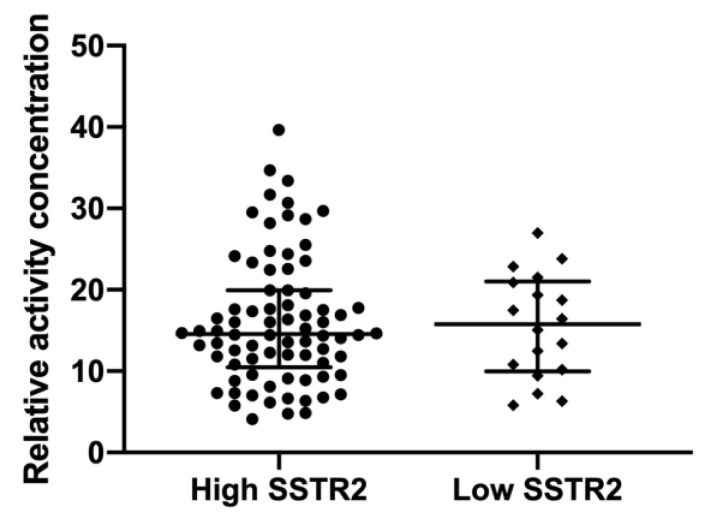
Relative activity concentration at SPECT 24 h after PRRT was estimated in 97 tumors from 33 patients: 27 with high and 6 with low SSTR2 expressing tumors. In each patient 2–3 tumors were measured, yielding 79 tumors in High SSTR2 and 18 tumors in Low SSTR2 groups, respectively. No significant difference was seen between the groups (*p* = 0.99, Mann–Whitney *U* test). Bars show median and inter-quartile range.

**Figure 8 cancers-13-02035-f008:**
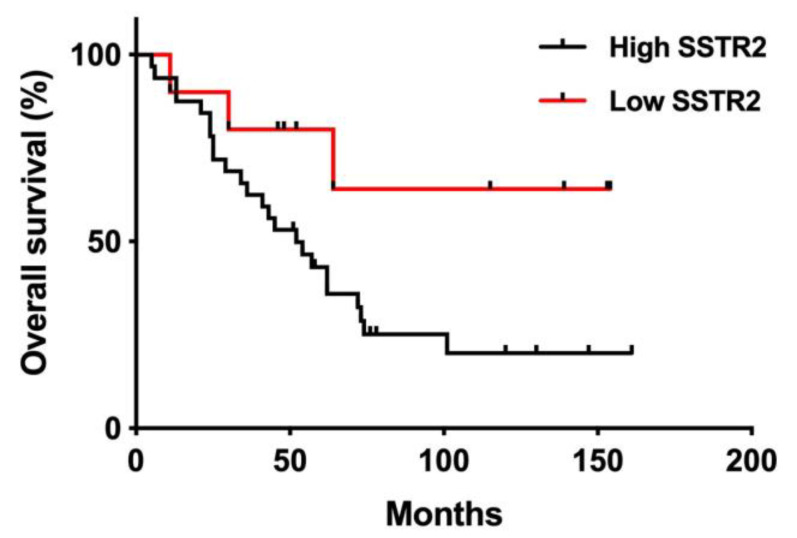
Overall survival of PRRT treated patients grouped according to SSTR2 expression (Low SSTR2, *n* = 10, High SSTR2, *n* = 32). Clinical characteristics of the patient groups are presented in Table 1. Patients in Low SSTR2 group had significantly prolonged OS compared with patients in High SSTR2 group. (*p* = 0.049; Mantel-Cox log-rank test).

**Table 1 cancers-13-02035-t001:** Clinical characteristics of PRRT treated patients.

	All Patients	High SSTR2 ^a^	Low SSTR2 ^b^	
Patients (*n*)	42	32	10	
Age at PRRT treatment, mean (range)	66 (45–78)	68 (48–77)	65 (45–78)	*p* = ns
Male/female (*n*)	19/23	15/17	4/6	*p* = ns
Number of PRRT treatments, mean (range)	3.8 (2–6)	3.8 (2–6)	3.7 (2–6)	*p* = ns
Ki-67%, median (range)	1.82	1.98 (0.25–15.2)	1.47 (0.62–3.1)	*p* = 0.10
Months between surgery and PRRT, median	60	62	53	*p* = 0.94

^a^ SSTR2 score 2 or 3 in all lesions; ^b^ SSTR2 score 0 or 1 in ≥1 lesion.

**Table 2 cancers-13-02035-t002:** Additional treatment after PRRT. None of the patients with low SSTR2 expression received further treatment except somatostatin analogues (SSA), after PRRT. HAE = hepatic artery embolization.

	All Patients	High SSTR2 ^a^	Low SSTR2 ^b^
Number of patients	42	32	10
Additional PRRT and/or HAE and/or external radiation	8	8	0
No additional treatment	34	24	10

^a^ SSTR2 score 2 or 3 in all lesions; ^b^ SSTR2 score 0 or 1 in ≥1 lesion.

## Data Availability

The data presented in this study are available on request from the corresponding author. The data are not publicly available due to ethical reasons.

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
