# Peer review of "Evaluation of SSTR2 Expression in SI-NETs and Relation to Overall Survival after PRRT"

_cancers, 2021, doi:10.3390/cancers13092035_

Round 1

Reviewer 1 Report

Overall this a well written and executed study- noting the small patient numbers and the time period over which patients had biopsies taken. Biomarkers apart from avidity on a SSTR-imaging to predict the efficacy of PRRT are urgently required.

The positives is the assessment of SSTR expression across tumoural deposits within patients, the correlation with the IMH markers that are utilised in evaluating NETs including Ki67 etc and uniform patient cohort i.e G and G2 Small Bowel NETs.

The conclusions are not surprising given the SSTR expression from a tissue specimen at a fixed time point would not be expected to correspond to the receptor dynamics and traffic and the pharmacodynamics of the radio-isotope in the NET cells. As stated by the authors the literature is inconsistent with heterogenous cohorts of patients. The limitation to low grade  small bowel NETs and the small patient numbers had been well accounted and are oblivious shortcoming of the analysis.

Nevertheless – I consider this as a very worthy addition to the literature.

Comments:

  1. Is there data on the stability of the SSTR staining with the age of the preserved embedded specimen?
  2. We argued that the presence of at least one tumor lesion with low SSTR2 expression implied a risk of having more lesions with low SSTR2 expression. Therefore, patients with at least one (1) tumor sample with score 0 or 1 were assigned to “Low SSTR2” group and remaining patients (all samples scored 2-3) were assigned to  “High SSTR2” group

Please justify this as we do see intrapatient tumoural heterogeneity based on SSTR imaging

  1. Comment on the other potential markers predicting the efficacy of PRRT. For example the NETest, post treatment imaging etc.
  2. Data on radiological or functional imaging response or PFS?
  3. Disease with low SSTR expression often with Grade 2 NETs do show concordant FDG avidity. The so-called sweet spot for PRRT in terms of deep responses and true prolongation of PFS/OS are often in these patients with grade 2 disease – often with concordant FDG avidity- whereby the radioisotope can express its true antiproliferative effects. Had any of these patients had FDG PET assessment?  

Reviewer 2 Report

The article by Elf and colleagues focuses on the retrospective assessment of SSTR2 expression in patients operated for small intestinal neuroendocrine tumors (SI-NETs) and subsequently receiving PRRT. The conclusion is that the retrospective evaluation of the levels of SSTR2 cannot be used to predict the overall survival (OS) of patients undergoing PRRT after surgical removal.

The study addressed a clinically relevant issue.  However, they reach a conclusion that contradicts several previously published articles on various NETs: they find no correlation between SSTR2 expression levels and response to PRRT.

As general comment, given that the entire study revolves around the IHC for SSTR2, the authors should also provide IHC pictures at higher magnification. It would be important to know where the signal is localized, cytoplasm or membrane, as the functional receptor is located in the membrane. This information should be included in Table 1. The localization of the signal is important, as it might in part explain the lack of an association between levels of SSTR2 and response to PRRT.

As the authors point out, a major caveat of the study is that the samples analyzed by IHC were collected many years before the patient underwent PRRT. Therefore, the study design was already somewhat flawed from the beginning as SSTR2 expression might have changed during that time.

Thus, the authors need to include in Table 1 the time when the samples were collected at surgery (years) and when the patients underwent PRRT. In Figure 1 they report the years when the samples spotted on the TMAs were collected, but in the current study only a small subset of these patients was analyzed and it is important to know how many years after samples collection the patients had PRRT.

Specific comments:

  • The samples selected for analysis were 44 patients for a total of 95 tissue samples. Two patients were excluded due to premature death and not included in the OS calculations. Given that the main goal of the study is to correlate SSTR2 expression with OS, I suggest to completely exclude these patients from the article. Consequently, the final number of samples analyzed is 32. This is to avoid confusion. There are already many numbers cited in the article, as every analysis they did included a different number of patients.
  • The paragraph starting from line 227 is not clear and seems to contradict what is indicated in the legend of Figure 4. Is there a difference in the level of SSTR2 between primary tumors and metastases? The text says that there is a difference, but in the figure is shown otherwise.
  • Figure 4. Some whiskers are not shown, why? It seems that something might be missing from the graphs.
  • I don’t much understand what is the meaning of the comparisons shown in Figure 4. The authors should provide the number of the samples included in each group, and then should explain why they did not simply compare the SSTR2 levels in primary and matched metastases but compared the “comparisons” between primary and mets.
  • Figure 5. Why the error bars for the IHC results for synaptophysin are not shown?
  • Line 255: I assume that the 66 samples analyzed for Ki67 derive from the 99 samples corresponding to the patients that received PRRT. This should be specified to help the reader understand the experimental setting.
  • The 2 panels in Figure 6 essentially show the same thing. So, it would be better to switch the position of the “Low SSTR2” and “High SSTR2” in the (b) panel so that from left to right both graphs go from lower SSTR2 expression to higher levels.

Also, please add a line indicating the 3% threshold in the graphs. To better understand which are the Grade 2 samples, they should be illustrated in color. This would help interpret the statement written in the figure legend that..” all but one samples with grade 2 tumors…”. As it is now, it is not clear which ones are these grade 2 samples and which one is the exception.

  • Line 290: How many patients were followed? All 42?
  • Line 291: what does the “7/10” in parenthesis refers to? Was not the total number of alive patients 15?
  • In figure 8 are depicted the 32 “high” and the 10 “low” samples? Please specify.

Reviewer 3 Report

The authors present an original study assessing the SSTR2 expression in SI-NETs as predicting OS after PRRT. This is an interesting study, however with few relevant limitations.

One of the major limitation of the study, is the long time between obtaining the samples (where SSTR2 staining were performed) and PRRT. Also, it is not clear to me the type of treatments that were received between this time period (which SSA, dosages, for how long, response to therapy, etc). Indeed, the authors mentioned in the paper that all patients underwent surgery from 1986 to 2013 which were then included in the TMA. So, there were very old samples used for IHC for SSTR2 and in such cases PRRT was given many many years later, so the tumor biology might have changed, as the authors mention in the discussion. This indeed could possibly explain some of the unexpected results, including lack of correlation between SSTR2 expression and 177Lu-DOTATATE uptake, or the poorer survival in patients with low SSTR2, which contradicts most of the published literature.

Could the authors add the time difference between obtaining tumor samples and PRRT treatment? At least, comparing this time difference between subgroups low vs high SSTR2? And describe more in detail the treatments given (like which SSA, dosage, period of treatment, response to treatment, etc). It is not clear in the methods which/how many patients have received SSAs and at timing of their disease course. E.g. there are any patients who received SSAs just before collection of the tumor specimen for IHC? The information regarding whether patients received pre-op SSAs should be mentioned. Also, the type of SSA should be mentioned as well, including as to whether any patient has received pasireotide?

Are there any relevant data from in vitro or in vivo studies that found similar findings of low SSTR2 expression correlating with poor survival?

Did the authors assessed other SSTR, like SSTR5 and SSTR1? Are there any studies correlating the expression of other SSTRs (other than SSTR2) with the DOTATATE uptake? Or with the overall survival of NETs?

Figure 5 aims to show correlation between SSTR2 expression and synaptophysin and CgA expressions. Wouldn’t be better to display these data with correlation plots and use Spearman correlation SSTR2vsSyn and SSTR2vsCrogA (assuming that the data is not normally distributed)? Instead of two-way ANOVA. Still about this figure 5, how do authors explain the lack of error bars in Synaptophysin data?

In figure 4, correct “metestasis” in the Y axis of the graph. Also a space should be given in “Lymphnode”

In the third paragraph of discussion about methodological considerations could the authors discuss the impact of using TMAs rather than full slides and how this could (if) influence the scoring of SSTR2? For the cases analysed, were SSTR2 stain homogeneous throughout the specimens or do the authors opt for selecting fields randomly or selecting hot spots? Could these methodological aspects explain the unexpected observations of this study? Further methodological descriptions regarding the assessment of IHC scores could be indeed provided in Methods section.

- The reference/cat no of the antibody anti-Ki67 (MIB1, Dako) has not been given. Please add it to the paper (Methods section)

- In introduction, Overall Survival (OS) abbreviation is explained with full expression twice (lines 67 and 79). To change to “OS” only in line 79. Same for TMA which is explained in lines 98 and 103.

- In methods section, line 132, please close brackets (after “group 3”)

Round 2

Reviewer 3 Report

The authors have satisfactorily addressed the points I previously raised. No further comments.